# Is the Canadian Healthy Eating Index 2007 an Appropriate Diet Indicator of Metabolic Health? Insights from Dietary Pattern Analysis in the PREDISE Study

**DOI:** 10.3390/nu11071597

**Published:** 2019-07-14

**Authors:** Jacynthe Lafrenière, Élise Carbonneau, Catherine Laramée, Louise Corneau, Julie Robitaille, Marie-Ève Labonté, Benoît Lamarche, Simone Lemieux

**Affiliations:** 1Institute of Nutrition and Functional Foods, Laval University, Québec, QC G1V 0A6, Canada; 2School of Nutrition, Laval University, Québec, QC G1V 0A6, Canada

**Keywords:** diet quality score, dietary guidelines, Healthy Eating Index, Alternative Healthy Eating Index, dietary pattern

## Abstract

The objective of this study was to identify key elements from the 2007 Canada’s Food Guide that should be included in a diet quality score aiming to reflect the risk of metabolic syndrome (MetS). Dietary intakes of 998 adults (mean age: 43.2 years, 50% women) were used to obtain the Canadian Healthy Eating Index 2007 (C-HEI 2007) and Alternative Healthy Eating Index 2010 (AHEI) scores, as well as a dietary pattern (DP) generated by the reduced rank regression (RRR) method. Based on these three scores, a modified version of the C-HEI 2007 (Modified C-HEI) was then proposed. The prevalence ratio (PR) of MetS was examined across diet quality scores using multivariate binomial regression analysis. A higher AHEI, Modified C-HEI, and a lower score for DP were all associated with a significantly lower prevalence of MetS (PR = 0.42; 95% confidence interval (CI) 0.28, 0.64; PR = 0.39; 95% CI 0.23, 0.63; and PR = 0.48; 95% CI 0.31, 0.74, respectively), whereas C-HEI 2007 was not (PR = 0.68; 95% CI 0.47, 1.00). Results suggest that a Modified C-HEI that considers key elements from the C-HEI 2007 and the AHEI, as well the DP, shows that participants with a higher score are less likely to have MetS.

## 1. Introduction

Canada’s dietary guidelines, presented in Canada’s food guide (CFG), were originally developed with the aim of ensuring the adequacy of the population’s macro and micronutrient intakes [1,2]. These guidelines are also aimed at promoting metabolic health, but the evidence supporting their ability to prevent chronic diseases is lacking. 

Metabolic syndrome (MetS) is characterized by a combination of at least three factors predicting chronic diseases, namely elevated blood pressure, fasting blood glucose, triglycerides, and waist circumference, and low high-density lipoprotein (HDL)-cholesterol concentrations [3]. It is a recognized measure of metabolic health. Indeed, a meta-analysis concluded that having this condition was associated with a two-fold increase in cardiovascular disease and a 1.5-fold increase in all-cause mortality [4]. Health-related habits, such as increased consumption of vegetables and fruits, whole grains, low-fat dairy, fish, legumes, and poultry, can reduce the risk of MetS [5]. Focusing on these factors in our dietary guidelines might, therefore, lead to improved metabolic health. 

Diet quality scores are often used to assess dietary habits. The Healthy Eating Index (HEI), first proposed in 1995 by Kennedy et al. [6], is intended to assess adherence to Dietary Guidelines for Americans. Since it was first introduced, three updated versions were published to ensure that the score remains coherent with evolving U.S. dietary recommendations [7]. Each version was successively validated [8,9,10] to demonstrate that it adequately measured adherence to dietary guidelines over the years. Through different studies, a higher HEI has been associated with reduced waist circumference [11], reduced risk of colorectal cancer [12], and reduced all-cause mortality [10,13], but only weakly with reduced cardiometabolic risk factors [14,15]. In line with recommendations included in the 2007 version of the CFG, a Canadian version of the HEI (C-HEI 2007) was proposed and validated [16]. Interestingly, Chiuve et al. [17] demonstrated that another score based on the scientific literature of metabolic disease prevention, the Alternative Healthy Eating Index 2010 (AHEI), was more strongly associated with the risk of coronary heart disease and type 2 diabetes than the American-HEI 2005. This suggests that some aspects of metabolic health might not be captured by the original HEI.

Both C-HEI 2007 and AHEI can be classified as a priori scores because they are based on predetermined components [18]. An increasing number of authors claim that these a priori scores are not optimal tools to promote healthy diet because they are not representative of the actual dietary intakes of the population [18,19]. Indeed, they do not take into consideration interactions between behaviour, food, and nutrients, and they may sometimes be difficult to translate into food-based dietary choices [20]. Therefore, there is a keen interest in using epidemiological approaches, such as dietary pattern analysis, to identify food habits associated with metabolic disease risk factors and to use this information to improve dietary guidelines for a specific population [20].

Besides the aim of improving metabolic health, dietary guidelines also aim to ensure sufficient intakes of the most important nutrients in a population’s diet and to limit overconsumption of nutrients of public health concern. Considering that the new version of the CFG has just been released [21], it is relevant to start updating measures of diet quality. Using comparison with other diet quality scores and dietary patterns derived from the reduced rank regression (RRR) approach, this study aimed to determine key elements from the 2007 CFG that should be included in a diet quality score for Canadians aiming to reflect the risk of metabolic diseases. More specifically, we first analyzed the C-HEI 2007 and the AHEI in order to identify reliable components for each of the two scores [22]. Second, we derived new dietary patterns that predict metabolic health and identified key components of these dietary patterns. Third, we used information from the first two steps to propose a modified version of the C-HEI 2007 that combines important elements of diet quality and that predicts metabolic health as reflected by the prevalence of the MetS.

## 2. Materials and Methods

### 2.1. Participants

This study was conducted with data from a large cross-sectional research project (PRÉDicteurs Individuels, Sociaux et Environnementaux (PREDISE) study) aiming to identify determinants of healthy eating in a probabilistic sample of French-speaking adults from the Province of Québec. Details about recruitment and collected measures are presented elsewhere [23,24]. Briefly, this multicenter web-based study was designed to examine how individual, social, and environmental factors are associated with the adherence to Canadian dietary guidelines. Participants were French-speaking men and women from 5 major administrative regions of the Province of Québec, Canada. To be eligible, participants had to be aged 18–65 years, speak French as the primary language at home, have a computer, have access to the internet, and have a valid e-mail address. This study was conducted according to the guidelines laid down in the Declaration of Helsinki and all procedures involving human subjects were approved by the Research Ethics Committees of Université Laval (ethics number: 2014-271), Centre hospitalier universitaire de Sherbrooke (ethics number: MP-31-2015-997), Montreal Clinical Research Institute (ethics number: 2015-02), and Université du Québec à Trois-Rivières (ethics number: 15-2009-07.13). Written informed consent was obtained from all subjects. From the 1216 participants of the PREDISE study, 998 were included in this analysis. Participants who did not have complete data for all of the required measurements (i.e., blood sampling, fulfillment of all three dietary recalls and all questionnaires; *N* = 214) or with type 1 diabetes (*N* = 4) were excluded.

### 2.2. Dietary Assessment and Participant Characteristics 

Participants had to complete three 24-h dietary recalls. The three days of recall were determined randomly and were completed over a 21-day period with a validated self-administered, web-based application, the Rappel de 24 h Web (R24W) [25]. This allowed the assessment of the mean intake of energy, macro and micronutrients, as well as the intake of different food groups necessary for the calculation of the C-HEI 2007 and the AHEI [26,27,28,29]. Scoring systems for both scores are described in Appendix A. The C-HEI 2007 is calculated on a total of 100 points, while the maximal score for the AHEI is 110 points. Both are obtained by the addition of different sub-scores, with higher total scores reflecting better-quality diets. Intake of 19 nutrients highlighted as key elements in the development of the 2007 CFG (vitamin A, thiamin, riboflavin, niacin, vitamin B6, folate, vitamin B12, vitamin C, vitamin D, magnesium, iron, calcium, potassium, sodium, zinc, phosphorus, fibres, linoleic acid and alpha-linolenic acid) [1] were assessed and compared with the average requirement or the adequate intake for age groups and genders [30]. 

Physical activity level was self-reported using a short version of the International Physical Activity Questionnaire (IPAQ) in metabolic equivalent (MET) minutes per week [31,32]. Information about social and demographic characteristics were obtained from online questionnaires. More precisely, we collected information about educational level, marital status, ethnicity and smoking status. 

### 2.3. Metabolic Health Measurements

Participants were invited to an initial visit at one of the affiliated research centres, where their body height (Height rod Model 216, SECA, SECA Corp, Hamburg, Germany), weight (TANITA body composition analyser, BC-418; Tanita Corporation, IL, USA), and waist circumference were measured. Waist circumference measurement was taken at the end of a normal expiration with a tape placed horizontally directly on the skin at the mid-distance between the last rib and the top of the iliac crest. Waist circumference was determined as the mean of three measurements to the nearest 0.1 cm. During this same visit, fasting blood samples (12-h fast) were collected from an antecubital vein into vacutainer tubes for the measurement of fasting blood lipids and glucose. Samples were then immediately centrifuged at 17 °C for 10 min at 1100× *g* and stored at −80 °C until processed. Blood lipid concentrations were assessed with the use of a Roche Modular P system (Roche Diagnostics, Mannheim, Germany). Fasting blood glucose concentrations were measured with the use of colorimetry (Hexokinase Method, Roche Modular P System), whereas insulin concentrations were tested with the use of electrochemiluminescence (Cobas 6000, Roche Diagnostics). Systolic and diastolic blood pressures were determined from the means of 3 consecutive measurements that were taken 3 min apart in a sitting position after a 10-minute rest with the use of an automated blood pressure monitor (Digital Blood Pressure Monitor HEM-907XL model, Omron, Toronto, Canada).

MetS definition is based on a consensus from the International Diabetes Federation and the American Heart Association/National Heart, Lung, and Blood Institute [3]. To be designated as having the MetS, participants had to have at least three of the following factors: (1) central obesity (waist circumference ≥ 88 cm for women and ≥ 102 cm for men); (2) triglycerides ≥ 1.7 mmol/L or self-reported specific treatment for high triglycerides; (3) HDL-cholesterol < 1.3 mmol/L for women and < 1.0 mmol/L for men or self-reported specific treatment for this lipid abnormality; (4) systolic blood pressure ≥ 130 mm Hg and/or diastolic blood pressure ≥ 85 mm Hg or self-reported treatment of previously diagnosed hypertension; or (5) fasting blood glucose ≥ 5.6 mmol/L or self-reported treatment of previously diagnosed type 2 diabetes. The definition from the Diabetes Federation and the American Heart Association/National Heart, Lung, and Blood Institute states that there is no universal cut point for waist circumference and that regional or national thresholds must be used. Therefore, we selected ≥ 88 cm for women and ≥ 102 cm for men, as they are cut-off points suggested for the Canadian population [33].

### 2.4. Dietary Pattern Development

Dietary patterns were obtained using reduced rank regression (RRR), a statistical technique that maximizes the variation explained by response variables selected based on the a priori hypothesis that they are related to the outcome of interest [18]. This approach shares similarities with the principal component analysis (PCA), but it is more appropriate to identify dietary patterns predictive of metabolic diseases [18]. Indeed, PCA are derived to explain as much variation as possible in food intake, while RRR describes the variation in response variables, which can either be a group of nutrients known as predictors or correlates of the outcome of interest, or biomarkers of this outcome [34]. For the development of dietary patterns in this study, biomarkers related to the MetS were used as response variables. We selected HDL-cholesterol, triglycerides, homeostasis model for insulin resistance (HOMA-IR), waist circumference, and mean arterial blood pressure, as they were either included in the calculation of the MetS or closely linked to it [34,35]. HOMA-IR was preferred to fasting blood glucose because the chronic increase in insulin resistance usually precedes that of fasting blood glucose, and HOMA-IR is slightly more closely associated with the MetS than fasting insulin [36]. Therefore, using HOMA-IR allows for an earlier identification of subjects prone to developing MetS [37]. HOMA-IR was obtained using glucose (in mmol/L) multiplied by insulin (in pmol/L) and divided by 135 [38]. Mean arterial blood pressure also seems to be a better predictor of the MetS in normotensive individual than systolic or diastolic blood pressure alone [39]. Based on the 2007 CFG, seventeen mutually exclusive food groups were used as predictors [1]: (1) dark green vegetables; (2) orange vegetables; (3) other vegetables; (4) whole fruit; (5) fruit juices; (6) whole grains; (7) refined grains; (8) yogurt; (9) milk; (10) other dairy products and alternatives; (11) nuts and legumes; (12) poultry; (13) eggs; (14) red and processed meat; and (15) fish and seafood. Food items not meeting criteria to be considered as a portion of the CFG were included as either: (16) sugar-sweetened beverages; or (17) “other foods’’ excluded from the CFG. Foods with a very high fat content (fats, dressings, mayonnaise, cream, etc.), foods with very high sugar content (sugars, honey, jams, syrups, sweets, etc.), foods high in fat, sugar, or salt (i.e., classified in the lowest level of diet quality (tier 4) according to the 2014 Health Canada Surveillance Tool Tier System [40]), and other ingredients and beverages (unsweetened beverages, sauces, condiments, etc.) were included in the “other foods” category. Details on the classification and portion sizes can be found in Appendix A. 

For the development of dietary patterns, we adjusted blood pressure levels in participants that reported taking any blood-pressure-lowering drugs by adding a correction constant (systolic blood pressure +15 mmHg; diastolic blood pressure +10 mmHg) [41]. Similarly, HDL-cholesterol and triglycerides were also adjusted in those taking lipid-lowering medication (triglycerides +0.208; HDL-cholesterol −0.060) [42], as suggested by Drake et al. As we used HOMA-IR, a variable derived using fasting insulin and fasting glucose, as one of the response variables, and since there was no adjusted criterion to account for the effect of the medication on HOMA-IR [43], we excluded all participants who self-reported using medication for diabetes (*N* = 38) for the RRR analysis. The score for the selected dietary pattern (DP) was generated by summing the product of standardized food group portion (*z*-score) coefficients and individual intakes for all participants [44].

### 2.5. Statistical Approaches

The C-HEI 2007 and AHEI scores were first assessed for internal consistency using the Cronbach-α. This statistical approach allowed determination of whether individual components of each score measure the same underlying construct. It is generally accepted that a Cronbach-α of ≥ 0.70 characterizes a score in which most of the components describe the same construct [45]. Furthermore, the analysis provides information about how each component independently correlates with the total score (comparing the component with the total score minus this component). For this analysis, correlation coefficients were categorized as moderate-strong (≥ 0.30) or null (< 0.10) [46]. In a second step, results from the RRR derived from all participants except those reporting taking diabetes medication (*N* = 960) were analyzed. Key elements from the DP were identified as those with factor loadings ≥ 0.20 [44] and were compared across quintiles of the distribution using general linear model. Based on these results, a modified version of the C-HEI 2007 (Modified C-HEI) was proposed by including only components of the C-HEI 2007 and AHEI that demonstrated a high internal consistency (components correlated with total score with coefficient *r* ≥ 0.30) or key elements identified in the DP (factor loadings ≥ 0.20). Prevalence ratio (PR) of MetS and each of its components were calculated across quintiles of C-HEI 2007, AHEI, DP score, and the Modified C-HEI using binomial regression analysis, while adjusting for age (in years, continuous), sex (female or male), physical activity levels (in MET minutes per week, continuous), energy intake (in kcal, continuous), smoking status (current smokers, current non-smokers), education levels (<college, college, university), marital status (single, married or in a couple), and ethnicity (Caucasian, other) in all participants (*N* = 998). All these co-variables were associated with at least one diet quality score or one marker of metabolic health in our sample. Adequacy of nutrient intakes associated with each score was evaluated using correlation analyses performed between each individual score and the cumulative number of nutrients (range 0 to 19) for which estimated average requirement or adequate intake was achieved per participant. Statistical analyses were conducted with the software Statistical Analysis System (SAS )version 9.4 (SAS Institute Inc., Cary, NC, USA) and comparisons of correlation coefficients were computed with MedCalc for Windows, version 15.0 (MedCalc Software, Ostend, Belgium) [47].

## 3. Results

### 3.1. Participants Characteristics

Participant characteristics are presented in Table 1. Women showed globally a better metabolic profile and higher dietary scores than men. However, men reported being more active than women. 

### 3.2. Internal Consistency of the A Priori Scores

Internal consistency of C-HEI 2007 and AHEI is described in Table 2. C-HEI 2007 demonstrated a better internal consistency than AHEI with a Cronbach-α of 0.61. Components with the strongest associations with the total C-HEI 2007 score were total vegetables and fruit, whole fruit, dark green and orange vegetables, whole grains, and “other foods”. Components not associated with the total C-HEI 2007 score were milk and alternatives and sodium. Similarly, the strongest contributors to the AHEI score were fruits and nuts and legumes, while those not associated with the total score were the sodium and alcohol components.

### 3.3. Dietary Patterns

Five dietary patterns were obtained from the RRR analysis. As the second, third, fourth, and fifth patterns explained < 10% of the variation in predictors (food groups) and less than 1% of the variation in the response variables (biomarkers), they were no further investigated. The DP derived from the RRR analysis is detailed in Table 3. It depicts foods inversely associated with HDL-cholesterol and positively associated with HOMA-IR, triglycerides, mean blood pressure, and waist circumference (Appendix A), and explained 11.5% of the variation in the predictors (food groups) and 3.8% of the variation in the response variables (markers of the metabolic health). Food groups with factor loadings ≥ 0.20 are presented as key food groups. DP was characterized by low intakes of whole fruits, orange and dark green vegetables, as well as yogurt, and high intakes of red and processed meat, sugar-sweetened beverages, refined grains, and “other foods”.

### 3.4. Modified C-HEI

The Modified C-HEI is presented in Table 4. Based on the internal consistency analysis (components showing a correlation with total score ≥ 0.30), components for vegetables and fruit, whole fruit, dark green and orange vegetables, whole grains, nuts and legumes, and “other foods” were kept, as they appear to be reliable elements of either the C-HEI 2007 or the AHEI scores. The other components were removed and replaced by new ones based on their factor loading from the dietary pattern analysis. Thus, we included yogurt (loading factor −0.20) as an adequacy component, as well as red and processed meat (loading factor 0.42), refined grains (loading factor 0.37), and sugar-sweetened beverages (loading factor 0.32) as moderation components. 

The suggested Modified C-HEI comprises 10 components, each having a maximal score of 10 for a total maximal score of 100 points (Table 4). We attributed the same maximal score for all components, as we had no information to justify a specific weighing of each of them [7]. To determine the scoring system for the proposed Modified C-HEI we decided that points for each component could be awarded following the logic of the C-HEI 2007 for all the items initially present in the score. For nuts and legumes, red and processed meat, and sugar-sweetened beverages, scoring from the AHEI was used. For the two newly added components (i.e., refined grains and yogurt), we had to suggest a scoring method. Based on the recommendation from the 2007 CFG that half of grain products should be whole grains, we suggested that intake of refined grains should be no more than half of the recommendation for total grain products. Therefore, anything above half of the recommended number of portions according to sex and age received zero points, and points were awarded proportionally for amounts between minimum and maximum. This approach is similar to the one used in the American-HEI 2010 [48] and in one Canadian adaptation of this score [49]. Finally, for the yogurt component, we determined the reference intake as 0.5 servings/day, as it was the mid-point between the average intake from the first and second quintile, where we noticed the most important modification in the distribution. Once again, points were awarded proportionally for amounts between minimum and maximum. The average Modified C-HEI score in our cohort was 44.2/100 (range 0.4–95.2). The Modified C-HEI was characterized by a Cronbach α of 0.68, which is slightly higher than each of the a priori scores tested. Moreover, all of its components were directly and significantly associated with the total score.

### 3.5. Prevalence of the Metabolic Syndrome

Prevalence of the MetS across quintiles of the four dietary scores is presented in Table 5. As the DP presented a reversed association with MetS, we used the fifth quintile (strongest adherence and lowest diet quality) as a reference instead of the first quintile. The first quintile was used as reference for the other scores (lowest adherence and lowest diet quality). The C-HEI 2007 was the only score not significantly associated with the prevalence of MetS (PR for comparison between fifth and first quintile = 0.68; 95%CI 0.47, 1.00; P for linear trend = 0.35). A higher AHEI and a lower DP score were associated with a lower prevalence of MetS (PR = 0.42 95% CI 0.28, 0.64 and 0.48; 95%CI 0.31, 0.74, respectively). 

The Modified C-HEI was strongly associated with the C-HEI 2007 (*r* = 0.76; *P* < 0.001) and the AHEI (*r* = 0.82; *P* < 0.001), while being negatively associated with the DP score (*r* = −0.68; *P* < 0.001), and was also associated with the MetS (PR for comparison between fifth and first quintile = 0.39 95% CI 0.23, 0.63). Appendix A details the comparison between the fifth and the first quintile of each of the four scores for individual MetS components. Overall, being classified in the quintile associated with the highest diet quality with all scores was associated with reduced prevalence of elevated blood pressure and elevated triglycerides, but only the Modified C-HEI was associated with a reduced prevalence of elevated blood glucose. All scores except C-HEI 2007 were associated with a reduced prevalence of increased waist circumference. Also, being classified in the quintile associated with the highest diet quality with AHEI was associated with reduced prevalence of low HDL-cholesterol. 

### 3.6. Adequacy of Nutrient Intake

Lastly, Table 6 illustrates nutrient adequacy across all the four diet quality scores. Scores for C-HEI 2007 and Modified C-HEI showed the strongest correlation with the cumulative number of nutrients for which estimated average requirement or adequate intake was achieved (respectively *r* = 0.42 and *r* = 0.36, *P* for comparison = 0.11), while the association was significantly lower for AHEI (*r* = 0.24) than for both C-HEI 2007 and Modified C-HEI (both *P* for comparison < 0.01). Meanwhile, the DP score was more weakly associated with the number of nutrients for which estimated average requirement or adequate intake was achieved than the three other scores (*r* = −0.05, *P* for comparison < 0.01). 

## 4. Discussion

This study aimed to identify components of the 2007 CFG on which we should focus to improve metabolic health. From analyses of a priori and a posteriori scores, we concluded that guidelines emphasizing the consumption of dark green and orange vegetables, whole fruit, whole grains, yogurt, as well as nuts and legumes, while suggesting to limit intakes of refined grains, red and processed meat, sugar-sweetened beverages, and food of low nutritional quality (i.e., other foods) would promote dietary habits associated with a lower prevalence of the MetS and a higher adequacy of nutrient intakes. Moreover, as opposed to the C-HEI 2007, and similarly to the AHEI, the Modified C-HEI was associated with the prevalence of MetS, while being more closely related to the adequacy of important nutrients than the AHEI. The Modified C-HEI also demonstrated a stronger internal consistency than both a priori scores. These results suggest that the Modified C-HEI would be an appropriate indicator of diet quality as defined by the adequacy of nutrient intake and metabolic health promotion.

Although most versions of the Canadian and American HEI scores have been widely used as measures of diet quality [50], our data and previous analyses suggest that their internal consistency are suboptimal [9,10,16]. Indeed, a recent validation of the American-HEI 2010 and 2015 demonstrated, similar to our results, that the dairy and sodium components are systematically inversely or not associated with the total score [9,10]. Although it is well established that a high sodium consumption leads to an increased risk of metabolic diseases [51], it has been shown that self-reported sodium is not necessarily an adequate marker of the real sodium intake [52]. Indeed, sodium intakes estimated using self-report dietary assessment methods are closely linked to sodium content of food items included in food databases, which could differ significantly from actual sodium content of foods currently available on the market. This could explain the discrepancies between sodium component score and diet quality scores, and it suggests that self-reported sodium intake is not a good marker of diet quality. Meanwhile, the recent literature highlighted either favourable or neutral associations between dairy products and metabolic health. Indeed, in a recent systematic review of meta-analysis of prospective populational studies, total dairy consumption demonstrated a favourable association with MetS with moderate-quality evidence, especially with regard to milk [53]. Moreover, a systematic review of randomized controlled trials revealed that the intake of yogurts containing Lactobacillus bulgaricus and Streptococcus thermophilus had either neutral or positive effects on markers of metabolic health [54]. The RRR analysis suggested that in our sample, milk was slightly associated with a deteriorated metabolic health (loading factor +0.15), while a higher consumption of yogurt (loading factor −0.20) was slightly associated with a reduced prevalence of MetS. In the new 2019 CFG, the food group that was dedicated to milk and alternatives has been integrated into the broader protein food group [21]. It is, therefore, likely that the updated version of the C-HEI 2007 based on the 2019 CFG will probably not specifically include milk and alternatives as a component of the score.

Since a C-HEI 2019 is not yet available, the C-HEI 2007 is currently the only available score based on Canadian dietary guidelines. Scores assessing adherence to American dietary guidelines are updated every 5 years and many changes have been proposed, such as a positive scoring for plant-based proteins and a negative score for refined grains in the American-HEI 2010 and 2015 [10,48]. This might explain why Liese et al. [13] demonstrated great similarities between the American-HEI 2010 and the AHEI in terms of population classification and mortality risk. With our cross-sectional data, we observed that the C-HEI 2007 was not associated with the prevalence of MetS, while the AHEI was associated with a significant reduction in the prevalence of the MetS between the fifth and the first quintile of score. However, the C-HEI 2007 was more closely linked to an adequate intake of important nutrients than AHEI. 

As expected, we observed that the DP derived with MetS-related biomarkers was highly associated with the MetS. This dietary pattern was characterized by a high intake of “other foods”, refined grains, sugar-sweetened beverages, and red and processed meat, and by a low intake of whole fruit, dark green and orange vegetables, as well as yogurt. The proportion of the variation explained in predictors (11.5%) and response variables (3.8%) may seem to be relatively small, but it is similar to what has been seen in other studies where biomarkers of the MetS were used as response variables [55,56]. For example, in a similar analysis conducted in Sweden, Drake et al. used the RRR and components of the MetS as response variables to derive a dietary pattern negatively associated with metabolic health that explained 3.2% of the variation in the response variables and 7.9% of the variation in the predictors [55]. The dietary pattern described as “Western” was characterized by large intakes of sugar-sweetened beverages, milk, artificially sweetened beverages, red and processed meat, as well as sweets and low intakes of wine, beer, cream, cheese, tea, and vegetables [55], which is relatively similar to the DP we obtained. In our sample, stronger adherence to the DP was associated with significantly higher prevalence of MetS. However, as opposed to the AHEI and the American-HEI 2015 [10,17], DP analysis suggests that an increase in the intake of fish and seafood (loading factor −0.15), as well as nuts and legumes (loading factor −0.15), are only weakly associated with a reduced prevalence of MetS. This could be explained in part by the relatively small contribution of fish and plant-based proteins to the diet of our participants. In fact, fish corresponded to 8% of all their portions of meat and alternatives, while nuts and legumes corresponded to 22%. In contrast, poultry, eggs, and red and processed meat represent 70% of meat and alternatives servings in our sample. As dietary patterns necessarily reflect actual dietary habits of the population in which analyses are conducted, they might minimize the importance of some diet-health associations in samples where these habits are not well implemented [34].

Compared to the C-HEI 2007, the Modified C-HEI scores refined grains and red and processed meat negatively. This explains in part why the average score is lower with the Modified C-HEI than with the C-HEI 2007. Similarly, the average score documented for the American-HEI 2010 was lower than the 2005 version of the score. The update from the 2005 to the 2010 version was characterized by the introduction of the negative scoring for refined grains [9]. Moreover, the Modified C-HEI attributes higher importance to plant-based components (vegetables and fruit, whole fruit, dark green and orange vegetable, whole grains, as well as nuts and legumes), which were those that had the highest association with the total C-HEI 2007 and AHEI scores, just as in the American-HEI 2010 and 2015 [9,10,17]. Suggesting increasing the consumption of non-refined plant-based food and to reduce red and processed meat, as well as highly processed food and sugar-sweetened beverages, is highly consistent with international consensus on diet, nutrition, and prevention of chronic diseases [57], and also with the recommendations of the new 2019 CFG [21].

### Strengths and Limitations

This study was conducted in a probabilistic sample of French-Canadians from the province of Québec. The sampling by sex and age groups was proportional to the population of each administrative region according to the most recent demographic data from the Institut de la statistique du Québec (2013) at the beginning of the study. A survey firm recruited participants by selecting phone numbers via random digit dialling until the designated quotas were reached in each stratum. This method ensures a good level of representativeness of the sample. 

Also, the RRR method is gaining popularity in epidemiological research as a modern approach to derive dietary patterns based on a priori hypothesis on relation between diet and health [58]. For our RRR analysis, we used a list of predictor food groups that was, however, shorter than what is usually seen in other studies. We wanted to use mutually exclusive subgroups of the CFG in order for the results to be meaningful [20]. In a similar analysis from Livingstone et al. [59] where 48 food groups were used as predictors, the dietary pattern associated with obesity was characterized by different types of vegetables and fruits, low-fat milk, and whole grains, while negative predictors were refined grains, processed meat, sugar-rich food, and beer and cider. These patterns are similar to the one we obtained using 17 food groups as predictors. 

When suggesting a new dietary score, it is challenging to ensure that all aspects are only data-driven. Therefore, selection and weighting of each component should reflect their relative importance. However, there is always some subjectivity involved in the process at different decision steps [7]. In the present study, we wanted to propose a new score that would be consistent with the 2007 CFG, therefore we kept the scoring based on the recommended number of portions by age groups and sex for most components. However, as the RRR analysis suggested that intakes of dark green and orange vegetables, whole fruit, as well as whole grains was highly important, we modified their contribution to make sure it was as important as all the other components. 

Data for this analysis were obtained from a validated web-based 24-h dietary recall. Three randomized days were used in order to average usual intake [27,28,29]. Some authors have suggested that a food frequency questionnaire is more appropriate to derive dietary patterns because it pictures food intake over weeks or months as compared to only a few days with dietary recalls [50,60]. However, it has been demonstrated that multiple days of 24-h dietary recalls produce energy and protein estimates that correlate better with biomarkers than those from food frequency questionnaires [52]. Moreover, the potential limitation induced by the small time scale assessed with 24-h dietary recall might be minimal in this case. Indeed, our data suggest that there is no difference in the average intake of the main 2007 CFG groups as well as energy and important nutrients across the four seasons in this cohort (*P* > 0.05, data not shown). On the other hand, these are cross-sectional observations, therefore we cannot input predictive value to our association. Moreover, the sample was slightly more homogenous in terms of ethnicity and more educated than the average population of Québec [61,62], therefore it is possible that a dietary pattern derived in a more representative cohort would have been slightly different. 

## 5. Conclusions

This study identified key elements from the 2007 CFG associated with metabolic health and adequate nutrient intakes in a cohort of French-speaking adults from the province of Québec. These results support public health campaigns that encourage Canadians to consume more vegetables, whole fruit, whole grains, yogurt, as well as nuts and legumes. Meanwhile, Canadians should be clearly advised to reduce their consumption of refined grains, red and processed meat, and highly processed foods high in saturated fat, sugar, and sodium. Indeed, a score based on these factors was associated with reduced prevalence of MetS. Interestingly, these factors are in accordance with the key concepts presented in the 2019 CFG [21] and might be included in a new version of the Canadian-HEI score reflecting current dietary recommendations. 

## Figures and Tables

**Table 1 nutrients-11-01597-t001:** Participants’ characteristics (*N* = 998).

	All Cohort (*N* = 998)	Women (*N* = 501)	Men (*N* = 497)	*P* ^1^
Age (year)	43.2 (13.5)	43.2 (13.5)	43.2 (13.4)	0.97
Body mass index (kg/m^2^)	27.4 (6.0)	26.9 (6.4)	27.8 (5.7)	0.03
Waist circumference (cm)	92.0 (16.3)	87.3 (15.4)	96.7 (15.8)	<0.01
Systolic blood pressure (mmHg)	117.7 (14.1)	112.8 (14.0)	122.6 (12.3)	<0.01
Diastolic blood pressure (mmHg)	73.6 (10.1)	72.0 (10.1)	75.2 (9.7)	<0.01
Triglycerides (mmol/L)	1.4 (0.9)	1.3 (0.9)	1.4 (1.0)	0.02
HDL-cholesterol (mmol/L)	1.4 (0.4)	1.5 (0.5)	1.3 (0.4)	<0.01
Fasting blood glucose (mmol/L)	5.2 (0.8)	5.1 (0.8)	5.3 (0.8)	<0.01
Fasting insulin (pmol/L)	98.2 (67.2)	94.2 (57.3)	102.2 (75.7)	0.06
HOMA-IR ^2^	3.9 (3.3)	3.7 (2.7)	4.2 (3.7)	0.02
Physical activity (METminutes /week) ^3^	3474.9 (5377.7)	2921.7 (4737.4)	4041.3 (5906.0)	<0.01
Average number of nutrients for which EAR or AI per person is achieved (/19) ^4^	13.8	13.9	13.6	0.07
Metabolic syndrome prevalence (%)	20.5	18.2	22.9	0.06
AHEI Score (/110) ^5^	53.5 (13.6)	56.6 (13.2)	51.4 (13.7)	<0.01
C-HEI 2007 Score (/100) ^6^	57.1 (14.1)	60.3 (13.0)	53.8 (14.4)	<0.01
Education (%)				
High school or less	23.8	23.4	24.3	
College	30.8	29.7	31.8	0.62
University	45.4	46.9	43.9	
Marital status (%) ^7^				
Single	35.6	38.6	32.6	0.05
In couple (or married)	64.4	61.4	67.4	
Ethnicity (%)				
Caucasian	92.8	92.7	92.9	0.65
Other ethnicity	7.2	7.3	7.1	
Smoking status (%)				
Current smoker	12.7	10.8	14.7	0.07
Non-smoker	87.3	89.2	85.3	

Data are presented as proportion or means (standard deviation). ^1^ Comparison between women and men: *T*-test for all variables except variables expressed in % where chi-square test is used. ^2^ HOMA-IR = homeostasis model for insulin resistance. ^3^ MET = Metabolic equivalent. ^4^ EAR or AI = estimated average requirements or adequate intake. ^5^ C-HEI 2007 = Canadian Healthy Eating Index 2007. ^6^ AHEI = Alternate Healthy Eating Index 2010. ^7^ Seventeen participants did not answer the marital status question.

**Table 2 nutrients-11-01597-t002:** Internal consistency (Cronbach-α) of the Canadian Healthy Eating Index 2007 (C-HEI 2007) and the Alternative Healthy Eating Index 2010 (AHEI) (*N* = 998).

	Internal Consistency (Cronbach-α)	Components Highly Associated with the Total Score (*r* ≥ 0.30)	Correlation with the Total Score (Excluding the Component)	Components Not Associated with the Total Score (*r* < 0.1)	Correlation with the Total Score (Excluding the Component)
C-HEI 2007	0.61	Vegetables and fruits	0.53	Milk and alternatives Sodium	0.08 −0.13
		Whole fruit	0.50		
		Dark green and orange vegetables	0.45		
		Whole grain	0.37		
		“Other foods”	0.43		
AHEI	0.49	Fruits	0.34	Sodium	0.001
		Nuts and legumes	0.31	Alcohol	0.06

**Table 3 nutrients-11-01597-t003:** Intake of key food groups (predictors) across quintiles (Q) of the selected dietary pattern (DP) (*N* = 998).

	Factor Loadings	Quintiles of DP ^1^										*P* for Trend
		Q1		Q2		Q3		Q4		Q5		
		Means	SD	Means	SD	Means	SD	Means	SD	Means	SD	
Red and Processed Meat (servings/day)	0.43	0.5	0.5	0.7	0.5	0.9	0.5	1.2	0.7	1.9	1.0	<0.001
Refined Grains (servings/day)	0.37	2.5	1.7	3.3	1.8	3.9	1.9	4.6	1.8	5.8	2.4	<0.001
Sugar-sweetened Beverages (servings/day)	0.32	0.1	0.4	0.2	0.4	0.4	0.7	0.5	0.9	1.4	1.7	<0.001
Other foods (kcal/day)	0.36	438.0	272.6	554.8	288.2	646.6	318.2	735.4	384.3	1053.6	580.2	<0.001
Whole Fruit (servings/day)	−0.34	2.4	1.6	1.5	1.1	1.1	1.0	0.9	0.9	0.7	0.8	<0.001
Dark Green Vegetables (servings/day)	−0.34	1.3	0.9	0.7	0.5	0.6	0.5	0.5	0.5	0.4	0.4	<0.001
Orange Vegetables (servings/day)	−0.27	0.8	0.9	0.4	0.4	0.4	0.4	0.3	0.4	0.3	0.4	<0.001
Yogurt (servings/day)	−0.20	0.5	0.6	0.3	0.4	0.3	0.4	0.2	0.4	0.1	0.3	<0.001

^1^ Only key food groups with a loading factor of ≥ 0.20 are presented.

**Table 4 nutrients-11-01597-t004:** Description of the modified version of the Canadian Healthy Eating Index Score (Modified C-HEI).

Component	Range of Scores ^1^	Scoring Criteria
Adequacy ^2^	0 to 60 points	
Total vegetables and fruit	0 to 10 points	Minimum: 0 Maximum: 7 to 8 servings
Whole fruit	0 to 10 points	Minimum: 0 Maximum: 1.5 to 1.7 servings
Dark green and orange vegetables	0 to 10 points	Minimum: 0 Maximum: 1.5 to 1.7 servings
Whole grains	0 to 10 points	Minimum: 0 Maximum: 3 to 4 servings
Yogurt	0 to 10 points	Minimum: 0 Maximum: 0.5 serving
Nuts and legumes	0 to 10 points	Minimum: 0 Maximum: 1 serving
Moderation ^3^	0 to 40 points	
Red and processed meat	0 to 10 points	Minimum: 0 Maximum: 1.5 servings
Refined grains	0 to 10 points	Minimum: 0 Maximum: 3 to 4 servings
Sugar-sweetened beverages	0 to 10 points	Minimum: 0 Maximum: 1 serving
“Other foods”	0 to 10 points	Minimum: 5% or less of the total energy intake Maximum: 40% or more of the total energy intake

^1^ Range of scores for adults. ^2^ For adequacy components, 0 points for minimum, 10 points for maximum or more, and proportional for amounts between minimum and maximum. ^3^ For moderation components, 10 points for minimum or less, 0 points for maximum or more, and proportional for amounts between minimum and maximum.

**Table 5 nutrients-11-01597-t005:** Prevalence ratio of the metabolic syndrome across quintiles (Q1 to Q5) of the Canadian Healthy Eating Index 2007 (C-HEI 2007), the Alternate Heathy Eating Index 2010 (AHEI), the selected dietary pattern (DP), and the Modified Canadian Healthy Eating Index score (Modified C-HEI) (*N* = 998).

	Mean Score (SD) Prevalence Ratio (95% CI)					
	Q1 ^1^	Q2	Q3	Q4	Q5	*P* for Linear Trend
C-HEI 2007 (/100)	36.9 (5.6) **1.00 (ref)**	49.1 (2.5) **0.84 (0.59, 1.19)**	57.6 (2.4) **0.83 (0.59, 1.18)**	65.4 (2.6) **0.77 (0.53, 1.10)**	76.3 (5.2) **0.68 (0.47, 1.00)**	0.35
AHEI (/110)	35.2 (4.8) **1.00 (ref)**	45.4 (2.3) **0.91 (0.66, 1.25)**	53.1 (2.1) **0.72 (0.49, 1.04)**	60.6 (2.5) **0.74 (0.53, 1.04)**	73.3 (6.1) **0.42 (0.28, 0.64)**	<0.001
DP ^2^	−0.59 (0.3) **0.48 (0.31, 0.74)**	−0.21 (0.1) **0.50 (0.34, 0.75)**	−0.01 (0.1) **0.86 (0.62, 1.21)**	0.20 (0.1) **0.76 (0.54, 1.08)**	0.61 (0.3) **1.00 (ref)**	<0.001
Modified C-HEI (/100)	19.7 (6.1) **1.00 (ref)**	33.9 (3.2) **0.85 (0.57, 1.26)**	44.1 (2.6) **0.59 (0.39, 0.91)**	54.0 (3.2) **0.69 (0.46, 1.05)**	69.3 (7.7) **0.39 (0.23, 0.63)**	0.003

Binomial regression adjusted for age (continuous), physical activity (continuous), energy (continuous), smoking status (2 categories), sex (2 categories), ethnicity (2 categories), marital status (2 categories), and education (3 categories). Note: *P* for linear trend across the prevalence ratio for all quintiles. ^1^ Reference is the 5th quintile (strongest adherence and lowest diet quality) for DP and the first quintile (lowest adherence and lowest diet quality) for C-HEI 2007, AHEI, and Modified C-HEI. ^2^ Scores for the DP were generated by summing the product of standardized food group portion (*z*-score) coefficients and individual intakes for all participants. The bold format aim at differentiate Prevalence ratio from mean score which are in the same boxes.

**Table 6 nutrients-11-01597-t006:** Correlation coefficients between the cumulative number of nutrients ^1^ for which estimated average requirement or adequate intake per person is achieved and each of the diet quality score (*N* = 998).

	C-HEI 2007	AHEI	DP	Modified C-HEI
Correlation coefficients (*P*)	0.42 ^a^ (<0.001)	0.24 ^b^ (<0.001)	−0.05 ^c^ (0.01)	0.36 ^a^ (<0.001)

Note: ^a, b, c^ = correlation coefficients with different superscript are significantly different (*P* for comparison < 0.01). C-HEI 2007 = Canadian Healthy Eating Index 2007; AHEI = Alternate Heathy Eating Index 2010; DP = selected dietary pattern; Modified C-HEI = Modified Canadian Healthy Eating Index. ^1^ The selected nutrients were: vitamin A, thiamin, riboflavin, niacin, vitamin B6, folate, vitamin B12, vitamin C, vitamin D, magnesium, iron, calcium, potassium, sodium, zinc, phosphorus, fibres, linoleic acid, and alpha-linolenic acid.

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
