# Peer review of "Is the Canadian Healthy Eating Index 2007 an Appropriate Diet Indicator of Metabolic Health? Insights from Dietary Pattern Analysis in the PREDISE Study"

_nutrients, 2019, doi:10.3390/nu11071597_

Reviewer 1 Report

I feel this paper adds to the current literature, the study is well done, and the methods and findings are adequately reported. I really enjoyed reading this paper and have no further comments. 

Author Response

July 3nd, 2019

Judy Hui Ph.D.

Managing Editor Nutrients

Dear Dr. Hui,

We were pleased to learn that our manuscript was potentially acceptable for publication in Nutrients. We have carefully revised the manuscript to take into account the points raised by the reviewers. You will find below a detailed description of the changes made following the reviewers’ suggestions and comments. All changes performed are clearly described in this letter and highlighted in the revised version of the manuscript. We are confident that these changes have improved our manuscript and hope that il will now be found of sufficient merit to be published in Nutrients. We look forward to your final decision.

Sincerely,

Jacynthe Lafrenière, Ph.D. Candidate

Institute of Nutrition and Functional Foods

Laval University

Reviewer 1

Thank you for the comments.

Reviewer 2 Report

This is a well written manuscript documenting how a modified dietary index was constructed and its comparison to other well-known dietary indexes. It provides a sound basis for a validation study.

My comments below are minor, and mainly related to the justification of choices or addition of detail.

L62 - "of the", not  "in the: most important nutrients..."

L63 - Sentence starting with "Knowing" is not at the same standard of writing as the others.

L65 - Could a justification of RRR over principle components analysis be added?

L93 - Can detail about how closely the 24hr recalls were collected be added? IE, consecutive, over 2 weeks, two weekdays, one weekend etc?

L158 - can explanation why those on HTN medication were not also dropped out of the analyses, as per the insulin medication? How many were on HTN medication?

L161-183. - If feel like some of these early analyses need to be included in the results section, especially since future decisions were based on them.

L163 - could a reference be added to the sentence on lines 162-164 about the method of summing the products.

Table 1: Not sure what the p values between men and women adds. They would be expected to be different in a number of ways. No further analysis of men and women is added later.

Discussion

Can a discussion and comparison about the results of the predictors and response variables be added to the discussion as to whether these are good results or not (Line 214-215).

Could more detail be added to Supplemental Table 1 in regard to the scoring system. I found some of them difficult to comprehend how some scores were given.

Author Response

July 3nd, 2019

Judy Hui Ph.D.

Managing Editor Nutrients

Dear Dr. Hui,

We were pleased to learn that our manuscript was potentially acceptable for publication in Nutrients. We have carefully revised the manuscript to take into account the points raised by the reviewers. You will find below a detailed description of the changes made following the reviewers’ suggestions and comments. All changes performed are clearly described in this letter and highlighted in the revised version of the manuscript. We are confident that these changes have improved our manuscript and hope that il will now be found of sufficient merit to be published in Nutrients. We look forward to your final decision.

Sincerely,

Jacynthe Lafrenière, Ph.D. Candidate

Institute of Nutrition and Functional Foods

Laval University

Reviewer 2

My comments below are minor, and mainly related to the justification of choices or addition of detail.

L62 - "of the", not  "in the: most important nutrients..."

Thank you for the comment, this has been modified as suggested (line 62).

L63 - Sentence starting with "Knowing" is not at the same standard of writing as the others.

We modified the sentence in order to improve the standard of writing. It now reads as follows: “Considering that the new version of the CFG has just been released, it is relevant to start updating measures of diet quality” (lines 64-65).

L65 - Could a justification of RRR over principle components analysis be added?

In this study, dietary patterns were obtained using reduced rank regression (RRR), a statistical technique that maximizes the variation explained by response variables selected based on the a priori hypothesis that they are related to the outcome of interest. This approach shares similarities with the principal component analysis (PCA), but it is more appropriate to identify dietary patterns predictive of metabolic diseases. Indeed, PCA are derived to explain as much variation as possible in food intake while RRR describes the variation in response variables which can either be a group of nutrients known as predictors or correlates of the outcome of interest or biomarkers of this outcome. We added details about our decision to select RRR instead of PCA in the materials and methods section of the manuscript (lines 137-143).

L93 - Can detail about how closely the 24hr recalls were collected be added? IE, consecutive, over 2 weeks, two weekdays, one weekend etc?

Days of completion of the 24-hour dietary recalls were selected randomly using an algorithm within the R24W software. The majority of the respondents (96.7%) completed the three recalls on non-consecutive days over a period of 21 days. Those details were added (line 93-96).

L158 - can explanation why those on HTN medication were not also dropped out of the analyses, as per the insulin medication? How many were on HTN medication?

Our cohort of 998 adults comprises 113 participants who reported taking blood-pressure-lowering medication. These subjects were not excluded from the development of the dietary pattern because we used adjustments suggested by Drake et al. (British Journal of Nutrition (2018), 119, 1168–1176) to take into consideration the effect of the medication on blood pressure. However, we did not find any published data about strategies to adjust insulin levels for taking into consideration diabetes medication. Therefore, we believe it was more rigorous to run the analysis of dietary patterns without subjects reporting taking medication for diabetes.

It is important to emphasize that although subjects on diabetes medication were excluded from the RRR analysis, all the 998 participants were included in the binomial regression analysis looking at the effect of the different dietary scores on the prevalence of the metabolic syndrome. This information was clarified in the materials and methods section (lines 181-182 and 187-192).

L161-183. - If feel like some of these early analyses need to be included in the results section, especially since future decisions were based on them.

We agree that the information concerning the selection of the dietary pattern based on the proportion of the variation in predictors and response variables explained should be in the results section. We made the change accordingly (lines 223-225).

L163 - could a reference be added to the sentence on lines 162-164 about the method of summing the products.

Thank you for the comment. The following reference should indeed have been added to the sentence: British Journal of Nutrition (2003), 89, 409–418. This addition has been made to the text (lines 173).

Table 1: Not sure what the p values between men and women adds. They would be expected to be different in a number of ways. No further analysis of men and women is added later.

We agree that there was not a strong focus on the sex differences in our analyses. However, considering that men and women were different on many aspects related to diet quality, we adjusted our binomial regression for sex. Therefore, we believe it is of relevance to include p values for the differences between men and women in Table 1.

Discussion

Can a discussion and comparison about the results of the predictors and response variables be added to the discussion as to whether these are good results or not (Line 214-215).

The proportion of variance  explained by the dietary pattern  in terms of  predictors (11.5%) and response variables (3.8%) may seem to be relatively small, but it is similar to what has been shown in other studies where biomarkers of the MetS were used as response variables. For example, in a similar analysis conducted in Sweden, Drake et al. used the RRR and components of the MetS as response variables to derive a dietary pattern negatively associated with metabolic health that explained 3.2% of the variation in the response variables and 7.9% of the variation in the predictors. This information was added in the discussion section as suggested (lines 359-364).

Could more detail be added to Supplemental Table 1 in regard to the scoring system. I found some of them difficult to comprehend how some scores were given.

A footnote as well as more information were added to the Supplemental table 1 to improve clarity of the scoring system that was used to calculate the Canadian Healthy Eating Index 2007.

Reviewer 3 Report

1. The authors state they used the metabolic syndrome (MetS) definition issued by the International Diabetes Federation (IDF) and the American Heart Association/National Heart, Lung, and Blood Institute, and indicate the following reference: Alberti, K.G.M.M.; Eckel, R.H.; Grundy, S.M.; Zimmet, P.Z.; Cleeman, J.I.; Donato, K.A.; Fruchart, J.-C.; James, W.P.T.; Loria, C.M.; Smith, S.C.; et al. Harmonizing the metabolic syndrome: a joint interim statement of the International Diabetes Federation Task Force on Epidemiology and Prevention; National Heart, Lung, and Blood Institute; American Heart Association; World Heart Federation; International Atherosclerosis Society; and International Association for the Study of Obesity. Circulation 2009, 120, 1640–1645. This international societies' statement is, without a doubt, the most recent and widely accepted variant for MetS definition. In this scientific statement, marked flexibility is seen when abdominal obesity is defined, leaving the cut-off values for waist circumference on the decision of each local authority. For example, Health Canada thresholds for waist circumference (also used by the authors of this manuscript: ≥ 88 cm in women and ≥ 102 cm in men) are usually identical to the United States values, but differ substantially from those recommended by the IDF in Europid subjects (≥ 80 cm in women and ≥ 94 cm in men), such as those dominant in Quebec area. Use of higher thresholds obviously selects a smaller number of MetS patients from a specific population. Trying to meet all opinions halfway, the IDF even suggests in the above-quoted statement that use of both pair of values would be helpful in studies on Europid populations. Therefore, I see it at the choice of the authors to either argue why they preferred to use only the higher cut-off values, or to double the statistical analysis and to extend the results and discussions by using both pairs of figures.

2. The authors referred to the values of fasting plasma insulin as a reference biomarker for MetS. What is their reason for not using HOMA indexes, which have the advantage of better reflecting the insulin resistance levels and can also be used in diabetes patients?

3. Exclusion of persons with drug-treated diabetes from the studied group (see above about the solidity of this exclusion criterion) may be perceived as a serious selection bias in a study trying to accurately identify MetS cases, since type 2 diabetes is frequently seen as a component of the MetS, and current guidelines do not advise for keeping patients with diabetes solely on lifestyle optimization. The authors should either repeat their analysis without eliminating medication-treated diabetes cases (i.e., using HOMA or other biomarker of the MetS that is still useful in the presence of diabetes and its dedicated drugs), or keep with their initial choice, but motivate it and state it as a limit of the study.

4. Minor typing errors (incorrect spacing between words) are seen at lines 3, 61, 136, 306-307, 315, 346, 351, 355,392, 417 and should be corrected.

Author Response

July 3nd, 2019

Judy Hui Ph.D.

Managing Editor Nutrients

Dear Dr. Hui,

We were pleased to learn that our manuscript was potentially acceptable for publication in Nutrients. We have carefully revised the manuscript to take into account the points raised by the reviewers. You will find below a detailed description of the changes made following the reviewers’ suggestions and comments. All changes performed are clearly described in this letter and highlighted in the revised version of the manuscript. We are confident that these changes have improved our manuscript and hope that il will now be found of sufficient merit to be published in Nutrients. We look forward to your final decision.

Sincerely,

Jacynthe Lafrenière, Ph.D. Candidate

Institute of Nutrition and Functional Foods

Laval University

Reviewer 3

1. The authors state they used the metabolic syndrome (MetS) definition issued by the International Diabetes Federation (IDF) and the American Heart Association/National Heart, Lung, and Blood Institute, and indicate the following reference: Alberti, K.G.M.M.; Eckel, R.H.; Grundy, S.M.; Zimmet, P.Z.; Cleeman, J.I.; Donato, K.A.; Fruchart, J.-C.; James, W.P.T.; Loria, C.M.; Smith, S.C.; et al. Harmonizing the metabolic syndrome: a joint interim statement of the International Diabetes Federation Task Force on Epidemiology and Prevention; National Heart, Lung, and Blood Institute; American Heart Association; World Heart Federation; International Atherosclerosis Society; and International Association for the Study of Obesity. Circulation 2009, 120, 1640–1645.

This international societies' statement is, without a doubt, the most recent and widely accepted variant for MetS definition. In this scientific statement, marked flexibility is seen when abdominal obesity is defined, leaving the cut-off values for waist circumference on the decision of each local authority. For example, Health Canada thresholds for waist circumference (also used by the authors of this manuscript: ≥ 88 cm in women and ≥ 102 cm in men) are usually identical to the United States values, but differ substantially from those recommended by the IDF in Europid subjects (≥ 80 cm in women and ≥ 94 cm in men), such as those dominant in Quebec area. Use of higher thresholds obviously selects a smaller number of MetS patients from a specific population. Trying to meet all opinions halfway, the IDF even suggests in the above-quoted statement that use of both pair of values would be helpful in studies on Europid populations. Therefore, I see it at the choice of the authors to either argue why they preferred to use only the higher cut-off values, or to double the statistical analysis and to extend the results and discussions by using both pairs of figures.

Thank you for the suggestion. The definition from the Diabetes Federation and the American Heart Association/National Heart, Lung, and Blood Institute states that there is no universal cut point for waist circumference and that regional or national thresholds must be used. Therefore, we selected ≥ 88 cm for women and ≥ 102 cm for men as they are cut points suggested for the Canadian population (Health Canada Canadian Guidelines for Body Weight Classification in Adults - Quick Reference Tool for Professionals Available online: https://www.canada.ca/en/health-canada/services/food-nutrition/healthy-eating/healthy-weights/canadian-guidelines-body-weight-classification-adults/quick-reference-tool-professionals.html (accessed on Jun 28, 2019)). This justification about the choice of waist circumference threshold values has been added to the manuscript (lines 131-135).

Using lower threshold values, as those representative of the European population (≥ 80 cm for women and ≥ 94 cm for men), would have increased the proportion of participants presenting a waist circumference higher than the cut point (42.7% vs 37.9%) but only slightly the proportion of participants categorized as having the metabolic syndrome (23.6% vs 20.5%).

2. The authors referred to the values of fasting plasma insulin as a reference biomarker for MetS. What is their reason for not using HOMA indexes, which have the advantage of better reflecting the insulin resistance levels and can also be used in diabetes patients?

We agree that HOMA-IR is probably a better predictor of the metabolic syndrome than fasting insulin, so we modified the RRR analysis and used HOMA-IR instead of fasting insulin (lines 144-151). As expected by the high correlation coefficient between HOMA-IR and fasting insulin (r=0.96 p<0.001) the dietary pattern derived was very similar to the one we derived initially. Almost all participants were classified in the same quintile of DP score using both methods (96.4%).

We agree that HOMA-IR can be used in diabetic patients to follow their level of insulin resistance. However, taking diabetes medication will have an impact on HOMA-IR values that is difficult to predict considering the different treatment regimens that exist for diabetes. Accordingly, we could not find a suggested adjustment method for diabetes medication as we found for medication used for blood pressure and blood lipids (Drake et al. (British Journal of Nutrition (2018), 119, 1168–1176). This is the reason why we have decided to conduct our RRR analyses without subjects on diabetes medication. 

Having said that, we would like to indicate to this reviewer that we have also conducted the RRR analysis using the entire cohort (including participant reporting taking medication for diabetes which corresponds to 3.8% of the cohort) and obtained very similar results. Indeed, the food pattern obtained had the exact same predictors with factor loading >0.20 (Red and processed meat, Other foods, Refined grains, Sugar-sweetened beverages, Green vegetables, Whole fruit, Orange vegetables and Yogurt). The proportion of variance explained in the responses variables was 4.1% and proportion of variance explained in the predictor was 11.3%. When the score derived with this food pattern was used in the binomial regression, being classified in the quintile representative of the highest diet quality was associated with a 0.48 prevalence ratio of having the metabolic syndrome as compared with being classified in the quintile representative of the lowest diet quality (this is highly similar to the data reported in the manuscript).    

3. Exclusion of persons with drug-treated diabetes from the studied group (see above about the solidity of this exclusion criterion) may be perceived as a serious selection bias in a study trying to accurately identify MetS cases, since type 2 diabetes is frequently seen as a component of the MetS, and current guidelines do not advise for keeping patients with diabetes solely on lifestyle optimization. The authors should either repeat their analysis without eliminating medication-treated diabetes cases (i.e., using HOMA or other biomarker of the MetS that is still useful in the presence of diabetes and its dedicated drugs), or keep with their initial choice, but motivate it and state it as a limit of the study.

We acknowledge that it was not clear in the original version of the manuscript that the sample used for the RRR analysis was not the same as the one used for assessing the prevalence of the MetS associated with the different dietary scores.   The participants who reported using diabetes-related drugs were excluded from the RRR analysis because we could not find a valid strategy to adjust for the effect of the medication on HOMA-IR values. However, when we classified all the participants as having or not the MetS in order to run the binomial regression analysis looking at prevalence ratios across the different scores, we included all participants. As mentioned in the materials and methods section (line 130-131), participant using diabetes-related medication were considered as having the factor for high blood glucose. This information was clarified in the materials and methods section (lines 181-182 and 187-192).

4. Minor typing errors (incorrect spacing between words) are seen at lines 3, 61, 136, 306-307, 315, 346, 351, 355,392, 417 and should be corrected.

Thank you for noticing those typing errors, They were all corrected as suggested.